# A Sensitive and Fast Fiber Bragg Grating-Based Investigation of the Biomechanical Dynamics of In Vitro Spinal Cord Injuries

**DOI:** 10.3390/s21051671

**Published:** 2021-03-01

**Authors:** Satyendra Kumar Mishra, Jean-Marc Mac-Thiong, Éric Wagnac, Yvan Petit, Bora Ung

**Affiliations:** 1École de Technologie Supérieure, 1100 Notre-Dame Street West, Montreal, QC H3C 1K3, Canada; satyendra-kumar.mishra.1@ulaval.ca (S.K.M.); eric.wagnac@etsmtl.ca (É.W.); Yvan.Petit@etsmtl.ca (Y.P.); 2Hôpital du Sacré-Cœur de Montréal, 5400 Gouin Boul. West, Montreal, QC H4J 1C5, Canada; jean-marc.mac-thiong@umontreal.ca

**Keywords:** fiber Bragg grating, spinal cord, strain, optical spectrum analyzer, photodiode

## Abstract

To better understand the real-time biomechanics of soft tissues under sudden mechanical loads such as traumatic spinal cord injury (SCI), it is important to improve in vitro models. During a traumatic SCI, the spinal cord suffers high-velocity compression. The evaluation of spinal canal occlusion with a sensor is required in order to investigate the degree of spinal compression and the fast biomechanical processes involved. Unfortunately, available techniques suffer with drawbacks such as the inability to measure transverse compression and impractically large response times. In this work, an optical pressure sensing scheme based on a fiber Bragg grating and a narrow-band filter was designed to detect and demonstrate the transverse compression inside a spinal cord surrogate in real-time. The response time of the proposed scheme was 20 microseconds; a five orders of magnitude enhancement over comparable schemes that depend on costly and slower optical spectral analyzers. We further showed that this improvement in speed comes with a negligible loss in sensitivity. This study is another step towards better understanding the complex biomechanics involved during a traumatic SCI, using a method capable of probing the related internal strains with high-spatiotemporal resolution.

## 1. Introduction

Characterization of spinal cord surrogates is important as it allows the study of internal strain sustained from spinal cord injuries. As a consequence, a quantitative description of their material properties, time-dependent stress-strain characteristics, and mechanisms of failure is a considerable undertaking. By measuring strain, we can quantify their elasticity and health, among others [1]. A number of techniques have already been proposed for the mechanical characterization, such as indentation probes [2], tension testers [3], compression techniques or rotatory shear techniques [4,5], and optical backscatter reflectometry [6]. All these techniques have common drawbacks, such as their inability to measure low-strain values and their slow response time.

In the last few decades, fiber Bragg gratings (FBGs) have being used extensively as sensors for the investigation of various parameters, such as temperature, pressure, displacement, humidity, strain, radiation, etc. [7,8,9,10,11,12,13,14,15,16,17]. FBGs consist of a periodic structure fabricated along the core of an optical fiber. In this periodic structure, the coupling of energy between different co-propagating and counter-propagating optical modes of the fiber takes place. The mode-coupling phenomenon is a strong function of the excitation wavelength. For a FBG written in a single-mode fiber, two identical counter-propagating modes are coupled, and the energy is transferred from the forward-traveling mode to the backward-traveling mode. The FBG, therefore, reflects certain wavelengths, thus acting as an optical bandpass filter [18].

For experimental spinal cord injuries (SCIs), the initial stage of spinal cord compression remains undetected with current sensing techniques, which prevents a comprehensive understanding of the spinal cord behavior under various loading conditions. Experimental replication of SCIs in vitro most often involves weight-drop contusion models. The contusion injury model is believed to be the most clinically relevant model of human SCIs [19]. Unfortunately, there is no method available for directly measuring the real-time biomechanics of contusion and mechanical behavior of the spinal cord [20]. This limitation strongly hinders our capability to study the relationships between the extent of the impact on the spinal cord, spinal cord loads/deformations, and functional outcomes.

Several studies have reported technologies capable of recording the spinal canal occlusion by providing indirect measurements of spinal cord compression. For example, a water-filled flexible polymer tubing simulating the spinal cord was inserted into the spinal canal for in vitro fracture replication [21,22]. A physical surrogate of the human spinal cord was also developed using silicone rubber that can be made radio-opaque. The same radio-opaque physical surrogate was used to simulate spinal canal occlusion with the help of a high-speed fluoroscope [23,24]. Unfortunately, these techniques are not able to provide any information about the measurement of internal forces and strains sustained by the spinal cord during the trauma.

Lucas et al. [25] proposed an imaging technique based on the fluoroscopic tracking of radio-opaque beads placed on the surface and within the spinal cord. However, fluoroscopic techniques are limited to a single plane of view (only in 2D), and visualization of the spinal cord is limited by the apparatus used to produce the SCI. Bhatnagar et al. [26] utilized MRI to quantify the internal deformation of the spinal cord, but this method cannot be used for the real-time assessment of impacts. The placement of the transducers on the spinal cord was restricted, and the long response time of the procedure precluded other methods of measurements [27]. Recently, we exploited the optical fiber bend loss method to evaluate the compression of a physical spinal cord surrogate based on light intensity modulation [28]. However, the latter method could not provide a precise localization of strain along the fiber (i.e., poor spatial resolution) and had a response limited to spinal cord compression rates larger than 40%.

In this work, we demonstrate an optical pressure-sensing scheme based on FBG technology that provides a highly sensitive strain measurement with cm-scale axial resolution and fast response (on μs scale) inside a surrogate spinal cord that is relevant to real-time in vitro studies of traumatic SCIs.

## 2. Model and Theory

### 2.1. Physical Spinal Cord Surrogate

The spinal cord surrogate was made of silicone elastomer foam (FlexFoam-iT III, Smooth-on, Macungie, PA, USA) and made by pouring the liquid primer form into a 3D-printed casting mold [29]. The dried foam acquired an ellipsoidal shape and the dimensions of 20 cm length, 11 mm major diameter, and 9 mm inner diameter (Figure 1) that mimic the human thoracic spinal cord transverse section. During the casting and curing of the silicone spinal cord surrogate, a 0.6 mm diameter nylon wire ran longitudinally through the surrogate. After curing, this wire was pulled out and replaced by the optical fiber inscribed with an FBG. Figure 1b depicts the transverse view of the mechanical and optical setup.

### 2.2. FBG Fiber Sensor Theory

A silica-based 125 micron diameter optical FBG pressure sensor (Technica LLC, Charleston, SC, USA) retains the intrinsic properties of Bragg gratings while offering the small size and mechanical compliance for insertion into a spinal cord surrogate. The procured FBG was written in a Ge-doped single-mode fiber at a length of about 3.5 mm and exhibited 90% peak reflectivity at the 1550 nm center wavelength. The silica glass cladding is coated with a polyimide polymer to protect it against humidity and scratches that can lead to crack formation and impairments to the fiber’s mechanical stability.

The sensing principle of the FBG is based on the Bragg reflection. The change in wavelength is caused by the axial strain change Δε and the temperature variation ΔT. The relation between temperature, wavelength, and strain is written as [25]
(1)$$\frac{\Delta \lambda }{\lambda }=\left(1-\;p_{e}\right)\Delta \varepsilon +\;\left(\kappa \;+\;\xi \right)\;\Delta T$$
where *λ* is the initial center wavelength of the FBG, Δλ is the shift in the Bragg wavelength, *p_e_*, *κ*, and *ξ* are the effective photo-elastic coefficient (for silica 4.22 × 10^−13^ (d/cm^2^)^−1^), the thermal expansion coefficient (8.52 × 10^−6^·K^−1^ at room temperature), and the thermo-optic coefficient (1.090 × 10^−5^/°C), respectively, of fused-silica fiber. The strain inside the FBG is defined as
(2)$$\Delta \varepsilon =\;\frac{\Delta \;L}{L}$$
where $\frac{\Delta \;L}{L}$ is the longitudinal strain applied on the FBG of length “*L*”. Note that in most cases of practical interest, the FBG is subjected to strains for which $\varepsilon_{x}=\varepsilon_{y}$, where these strains are in *x* and *y* directions. It is also assumed that a standard Poisson relation relates the transverse strains to the longitudinal strain, i.e., $\varepsilon_{x}=\varepsilon_{y}=-\nu \varepsilon_{z}$ [30].

If in Equation (1) we consider Δ*T* = 0 (constant temperature), then the equation simplifies to
(3)$$\frac{\Delta \lambda }{\lambda }=\left(1-p_{e}\right)\Delta \varepsilon .$$

Note that the photo-elastic coefficient of the fiber core (*p_e_*) can be expressed as
(4)$$p_{e}=\frac{n_{eff}^{2}}{2}\left[p_{12}-v\left(p_{11}+p_{12}\right)\right]$$
where *p*_11_ and *p*_12_ denote the photo-elastic tensor components (typical values for fused silica: *p*_11_ = 0.113 and *p*_12_ = 0.252), *v* is Poisson’s ratio (typically for fused silica: *v* = 0.17), and *n_eff_* is the effective index of the core-guided optical mode. Typical values of the coefficients in silica-core optical fibers yield *p_e_* = 0.21, resulting in a strain sensitivity (Δ*λ_B_/ε*)~1.2 pm/με and strain limits of +/− 2500 με.

## 3. Results

### 3.1. Characterization of FBG Sensor with OSA

First, optical characterization of the FBG sensor was performed using an optical spectrum analyzer (OSA). A schematic diagram of the setup is shown in Figure 2. We used a broadband near-infrared source (EDFA, 1520–1560 nm), optical circulator, FBG, and OSA. We connected the FBG sensor to the laser source through the circulator, while the other arm of the circulator was connected to the OSA for detection. The broadband light was coupled to the FBG fiber sensor via the optical circulator, which also collects the back-reflected signal from the sensor that is then measured by the OSA. Stretching the fiber causes a minute change in the grating period that shifts the center Bragg wavelength in the reflection spectrum. Therefore, before recording any data, we fixed the instrumented spinal cord (with the FBG sensor) with glue on a glass slide. The weight pressure was applied in the normal direction in the middle of the spinal cord for 4 min and then the measurement was recorded. Next, we changed the weight (3.34, 7.97, 9.78, and 24.98 g) and recorded the data. The diameter of the different weights applied was kept constant at 1 cm. Without applying any weight, the full-width half-maximum (FWHM) bandwidth was calculated from the recorded reflectance spectrum as 0.56 nm, and the peak wavelength was 1549.88 nm.

Figure 3 depicts the reflectivity curve for different weight loads over the spinal cord attached with the FBG. We observed that as we increased the weight over the spinal cord, the reflectivity spectrum redshifted (Figure 3), as expected from theory and owing to strain-induced local modification in the grating period of the FBG. From the results, it is clear that as we increased the pressure, the resonance wavelength redshifted (Figure 4a), with a corresponding change in the local strain (as calculated with Equation (3)), and exhibited a 7% relative error in the linear curve fit displayed in the plot. Since the optical bandwidth was relatively large (0.56 nm), it limited the accuracy in determining the value of the wavelength corresponding to the peak power. Figure 4b shows the change in the local strain with respect to the shift in the resonance wavelength.

### 3.2. Calibration of Compression Sensing Capability of Spinal Cord

Figure 5 depicts the setup used for evaluating the compression sensing capabilities of the proposed sensing scheme, where we can see the cylindrical steel impactor that was used to compress the spinal cord surrogate, mounted on a three axis-stage with micrometer positioners. In Figure 6, we plotted the change in the optical power as a function of transverse compression applied to the spinal cord surrogate. Optical power was calculated as a percentage of its initial value, measured at the beginning of the first load cycle (with no load applied). The procedure was performed five times in order to ensure repeatability. In Figure 6, we plotted the microstrain induced by the transverse compression (as deduced from the optical power change) and observed, as expected, an increase in the measured strain with the compression rate. Notably, it was shown that even though the FBG sensor was buried 1 mm below the surface of the spinal cord surrogate, the optical sensing scheme exhibited a quasi-immediate response to compression rates over 0%. On further increase of the transverse loading, the sensor showed a good response, and in particular, inside the range of 20–60% compression, a quasi-linear relationship was observed between the measured microstrains with respect to the applied transverse compression. We note that the latter results represent a significant improvement in the detection limit over the previous method employing fiber bend losses [29], where a minimum 40% compression was required for the onset of the optical sensor response.

### 3.3. Fast Transverse Compression Measurement Using a Narrow-Band-Pass Filter and Power Meter

Further, we interfaced the circulator with a power meter via a narrow-band-pass filter that reduced the bandwidth of the reflection spectrum, and therefore allowed for a more accurate measurement of the wavelength shift Δλ. The FWHM of the ensuing reflection spectrum was measured at 0.11 nm (i.e., one-fifth of the previously unfiltered spectra). Figure 7 shows the schematic diagram of the modified setup where, notably, we replaced the OSA with a power meter.

As is well known, the response time of a typical OSA is slow (around 2 s). In order to record fast events such as a SCI, we replaced the OSA with a fast Si photodiode-based optical power meter connected to a digital interface for data acquisition with a response time of 20 μs. The response time thereby improved by five orders of magnitude compared to the conventional OSA-based method.

Figure 8a shows the variation of power with respect to the applied pressure as well as the linear curve fit that exhibited a 3% relative error. As indicated previously, the applied pressure induces transverse compression in the FBG section, which in turn redshifts the center wavelength of the reflection spectrum. As the reflection spectrum passes through the narrow-band filter, the decrease in peak intensity directly depends on the applied pressure, as shown on Figure 8a. A transverse compression variation curve, when fitted linearly, shows a relative error of 8.9%. Since the change in Bragg wavelength is directly proportional to the change in strain (transverse compression), we can thus calculate the transverse compression of the FBG fiber (assuming a constant temperature).

Figure 8b shows the variation of the transverse compression sustained by the surrogate spinal cord with respect to changes in optical power loss that indicates a quasi-linear relation. The latter curve allows us to define and evaluate the sensor’s sensitivity in terms of transverse compression as a function of relative power loss, at 10.3% (which in decibel units translates into 0.9% transverse compression/dB optical power loss), a fairly sensitive value to be expected for FBG-based strain sensors.

## 4. Discussion

Considering that during a SCI involving a burst-type fracture, bone fragments can be propelled through the spinal cord at speeds of 4.5 m/s [31], the injury timescale is around 2 ms. In this paper, we proposed and demonstrated a fiber optic FBG strain sensor with a fast response time of around 20 μs, thus allowing the full and accurate temporal investigation of strain dynamics occurring in a spinal cord surrogate for the in vitro replication of SCIs. All the while, the proposed scheme maintains the compact size, unobtrusiveness, and mechanical compliance that are hallmark features of optical fiber sensors. The proposed method moves away from the slow and costly OSA, traditionally used for performing spectral measurements of the FBG sensor response; instead, this method uses a static and cost-effective narrow-band filter to obtain power loss measurements that are correlated to the spectral shift in the FBG’s peak reflectivity (and ultimately the internal strain values). The latter method’s speed enhancement of over five orders magnitude, gained by replacing spectral interrogation with power measurements, was obtained with a small loss in sensor accuracy in the process. We note that the latter accuracies can be further improved by adding more sample measurements in the calibration curves. Owing to the high sensitivity of the FBG-based optical sensor and the fast (microsecond scale) response of the proposed power sensing scheme, we believe that this method provides a new tool to further investigate the real-time strain and compression dynamics that occur during SCI events, using artificial or cadaveric spinal cord samples instrumented with the FBG optical fiber sensors.

## Figures and Tables

**Figure 1 sensors-21-01671-f001:**
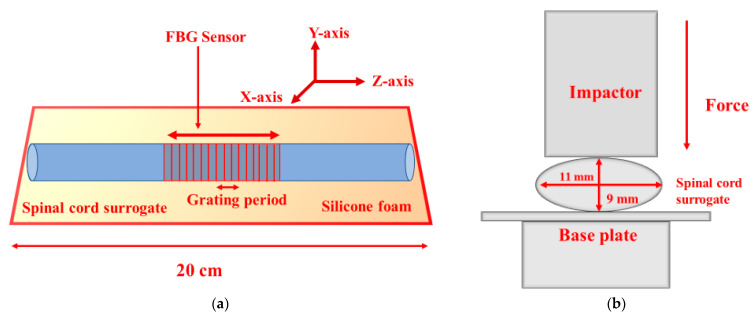
(**a**) Schematic representation of the longitudinal mid-cross-section of the instrumented spinal cord surrogate with the integrated fiber Bragg grating (FBG) sensor. (**b**) Transverse view of the mechanical and optical setup.

**Figure 2 sensors-21-01671-f002:**
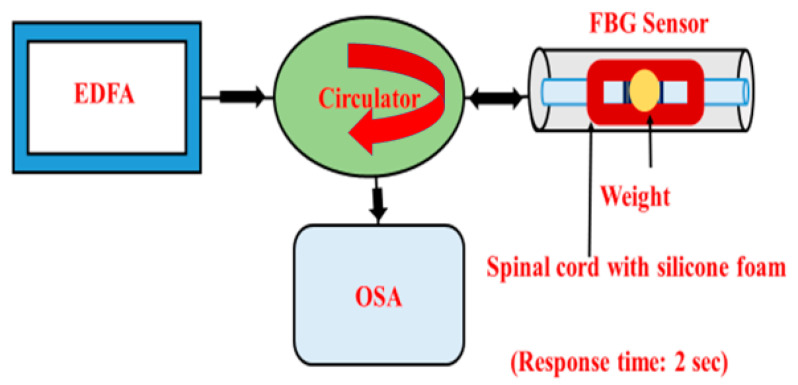
Experimental setup for the characterization of the fiber Bragg grating FBG sensor.

**Figure 3 sensors-21-01671-f003:**
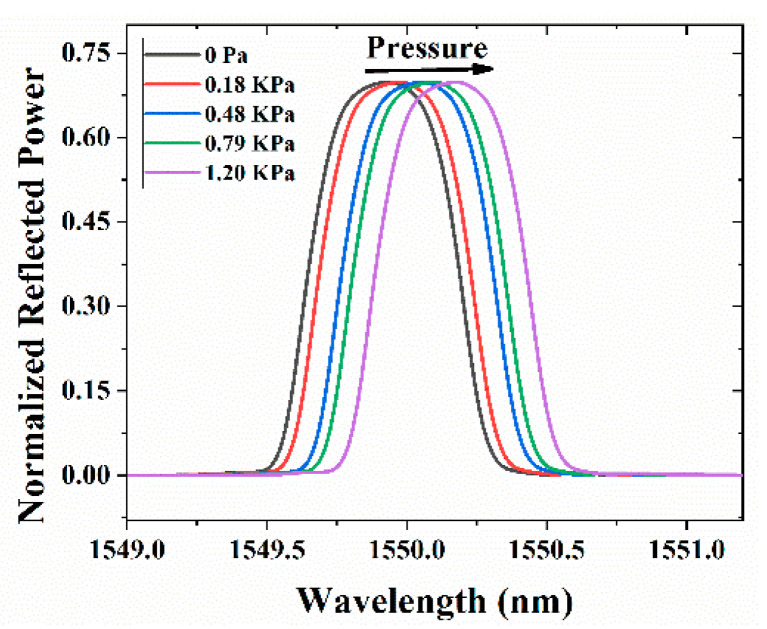
Reflection spectra (normalized with input power) of the sensor for different transverse pressures on the spinal cord surrogate using the optical spectrum analyzer.

**Figure 4 sensors-21-01671-f004:**
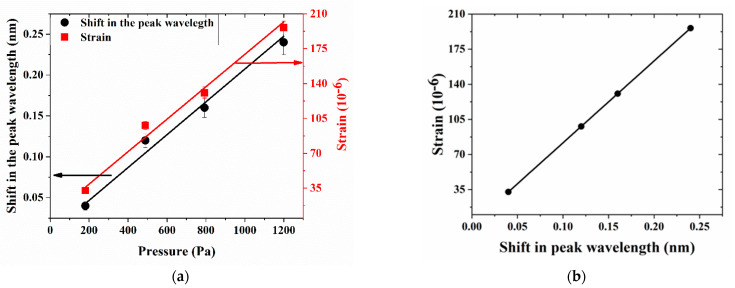
(**a**) Variation of peak wavelength and strain with respect to applied pressure. (**b**) Sensor relationship between the strain as a function of the peak wavelength shift.

**Figure 5 sensors-21-01671-f005:**
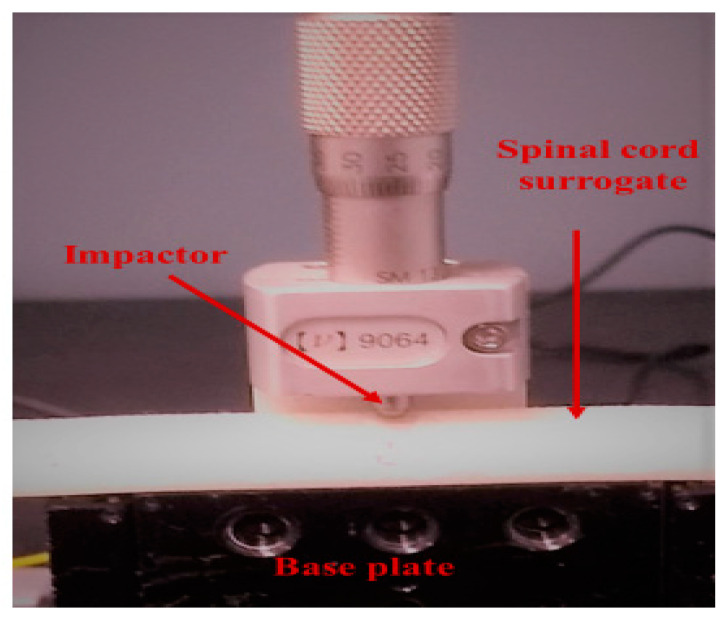
Instrumented spinal cord surrogate under testing.

**Figure 6 sensors-21-01671-f006:**
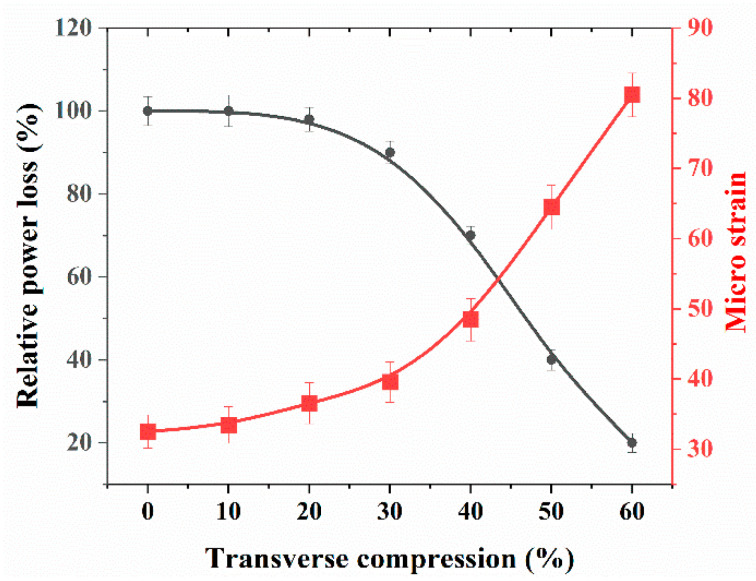
Relative power loss and microstrain plotted as a function of spinal cord transverse compression for static loading.

**Figure 7 sensors-21-01671-f007:**
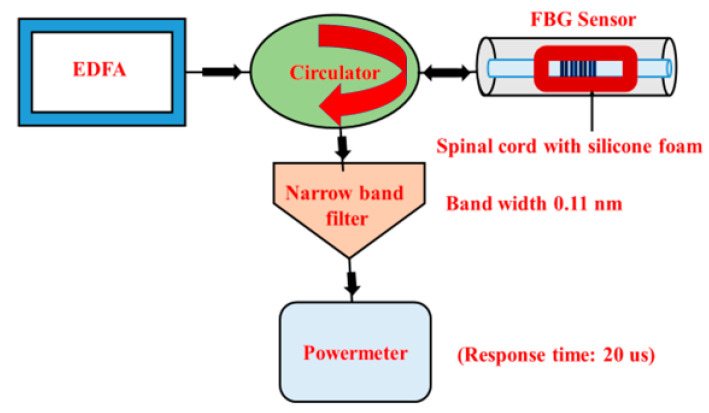
Experimental setup of the strain FBG sensor based on a fixed narrow-band optical filter and a fast photodiode (i.e., power meter).

**Figure 8 sensors-21-01671-f008:**
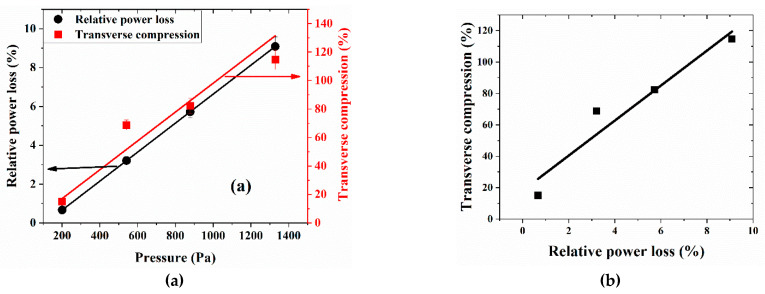
(**a**) Variation of power loss and transverse compression with respect to applied pressure. (**b**) Sensor relationship between the transverse compression as a function of the relative power loss.

## Data Availability

This study does not report any additional public data.

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
