# Peer review of "A Sensitive and Fast Fiber Bragg Grating-Based Investigation of the Biomechanical Dynamics of In Vitro Spinal Cord Injuries"

_sensors, 2021, doi:10.3390/s21051671_

Round 1
Reviewer 1 Report
In this paper, the FBG technique was used to test the transverse compression of the spinal cord model in order to study the changes of the spinal cord in ruptured fractures. This article is compact in structure, detailed in content, and has considerable novelty.
The following suggestions are for the author's reference:
- In this paper, the author says “Unfortunately, there is no method available for directly measuring in real time the biomechanics of contusion and mechanical behavior of the spinal cord”. So how to verify the author's test results using FBG sensors are correct? So how to verify the author's test results using FBG sensors are correct? Is it possible to compare the theoretical results with the measured results?
- There are some formatting problems, such as inconsistent indentation at the beginning of a paragraph, unindented paragraphs (line 31, line 40, etc.), excessive indentation in some paragraphs (line 71, line 171, etc.);the formula is not centered; and so on. It is suggested to modify it.
- Figure 1 and text do not correspond perfectly. It is suggested in Figure 1 that all dimensions be clearly marked, i.e., "20cm length, 11 mm major diameter and 9 mm inner diameter" as shown in the article. It is also recommended to redraw the FBG sensor in Figure 1.
- In Section 2.1, “The spinal cord surrogate was made of silicone elastomer foam (FlexFoam-iT III, Smooth-on, PA, USA) made by pouring 76 the liquid primer form into a 3D printed casting mold.” Does the material mimic the spinal cord well? How to ensure that the fiber sensor and the model deformation in harmony? In addition, it is recommended to give the actual picture of the model.
- The description of the test process in this paper is too brief, and it is difficult to fully show the test process in Figure 2. It is suggested to give a more detailed description and add photos of the test process.
- Some pictures are not clear enough to fully display, such as Figure 5.It is recommended that they be checked and modified.
- In Section 2.2, it is recommended to give the specific values of parameters κ, ξ, etc
- Some references have problems, for example, references.[2][3][4], etc, and it is recommended to check.
- The subtitle is not representative. For example, Section 2.2 should be the principle of FBG sensors, etc. It is suggested to be modified.
To sum up, I suggest this article can be accepted.

Author Response
In this paper, the FBG technique was used to test the transverse compression of the spinal cord model in order to study the changes of the spinal cord in ruptured fractures. This article is compact in structure, detailed in content, and has considerable novelty.
The following suggestions are for the author's reference:
- In this paper, the author says “Unfortunately, there is no method available for directly measuring in real time the biomechanics of contusion and mechanical behavior of the spinal cord”. So how to verify the author's test results using FBG sensors are correct? So how to verify the author's test results using FBG sensors are correct? Is it possible to compare the theoretical results with the measured results?
Response: It is difficult to predict actual deformation and mechanical behavior of the spinal cord with arbitrary accuracy. The biomechanical behavior of a person’s spinal cord varies from patient to patient as well as the precise location of the contusion. But on the other hand, FBG based fiber optics sensing is a well-established method with high sensitivity and reproducibility. We believe that using this tool we can calculate the desired parameters with high degree of accuracy.
- There are some formatting problems, such as inconsistent indentation at the beginning of a paragraph, unindented paragraphs (line 31, line 40, etc.), excessive indentation in some paragraphs (line 71, line 171, etc.);the formula is not centered; and so on. It is suggested to modify it.
Response: We thank the reviewer for indicating these issues. We have modified and corrected formatting issues in the revised version of the paper as per the suggestions.
- Figure 1 and text do not correspond perfectly. It is suggested in Figure 1 that all dimensions be clearly marked, i.e., "20cm length, 11 mm major diameter and 9 mm inner diameter" as shown in the article. It is also recommended to redraw the FBG sensor in Figure 1.
Response: We have modified and redrawn figure 1 as per the Reviewer’s suggestions.
- In Section 2.1, “The spinal cord surrogate was made of silicone elastomer foam (FlexFoam-iT III, Smooth-on, PA, USA) made by pouring 76 the liquid primer form into a 3D printed casting mold.” Does the material mimic the spinal cord well? How to ensure that the fiber sensor and the model deformation in harmony? In addition, it is recommended to give the actual picture of the model.
Response: Yes, the material and the casted spinal cord does mimic a human spinal cord quite accurately. This aspect was studied in detail in a prior study: L. Diotalevi, Y. Petit, L.-M. Peyrache, Y. Facchinello, J.-M. Mac-Thiong and E. Wagnac, “A novel spinal cord surrogate for the study of compressive traumatic spinal cord injuries”, Proc. of EMBC, July 2019. We have added the corresponding reference of this work [XX] in the manuscript so as to provide additional details to prospective readers.
- The description of the test process in this paper is too brief, and it is difficult to fully show the test process in Figure 2. It is suggested to give a more detailed description and add photos of the test process.
Response: As per reviewer’s suggestion we have added a sentence in Section 3.1 to explain the setup: “The broadband light is coupled to the FBG fiber sensor via the optical circulator, which also collects the back-reflected signal from the sensor that is then measured by the OSA.” We have also modified Fig. 2 with an arrow sign on the circulator (as done typically) to more explicitly indicate its function.
- Some pictures are not clear enough to fully display, such as Figure 5. It is recommended that they be checked and modified.
Response: As per reviewer’s suggestion we have checked all figures and improved the readability whenever necessary. In particular, we added a schematic in Fig. 1(a) to clearly show the mechanical and optical setup used in the experiment.
- In Section 2.2, it is recommended to give the specific values of parameters κ, ξ, etc
Response: As recommended, we have added the specific values of κ, ξ in the revised manuscript.
- Some references have problems, for example, references.[2][3][4], etc, and it is recommended to check.
Response: We thank the Reviewer for noticing this formatting issue. We have thoroughly checked the references in the manuscript and corrected them as per the journal’s format.
- The subtitle is not representative. For example, Section 2.2 should be the principle of FBG sensors, etc. It is suggested to be modified.
Response: We have modified the subtitle in Section 2.2, as per the Reviewer’s suggestion, for: “FBG Fiber Sensor Theory”.

Reviewer 2 Report
This paper (Sensors-1078234) presents an optical pressure-sensing scheme based on a fiber Bragg grating (FBG) for detecting the transverse compression inside a vitro spinal cord. Due to the inherent property of FBG of high sensitivity to strain measurement, it would be a potential method to investigate the traumatic spinal cord injury (SCI). By utilizing the inherent high-speed response of the Si photodiode-based optical power meter, the response time of the measurement can be improved greatly. The paper gives the fundamental research of the FBG based sensor for traumatic SCI. It is a good exploration by using optical sensors to monitor the real-time biomechanics of soft tissues. The following should be addressed properly before it can be recommended to Sensors. (1) The narrow band filter is utilized in Figure 7 to obtain power loss measurements that are correlated to the spectral shift in the FBG’s peak reflectivity. The relationship between the central wavelength of the narrow band filter and reflection spectrum the FBG should be analyzed. To obtain high precision of power loss measurement, the bandwidth of 0.11 nm is relatively large, which should be narrowed further greatly. (2) In Figure 6, optical power is plotted as a function of spinal cord transverse compression for static loading. How about the repeatability? (3) Figure 3 presents the reflection spectra of the FBG based transverse pressures on the spinal cord surrogate. The unit of Y-axis is dBm and the contrast ratio is only about 0.68 dB. The reflection ratio is low. The unit of dBm should be checked. It may be the absolute value of optical power. (4) In page 6, line 177-178, “As the reflection spectrum passes through the 177 narrowband filter, the decrease in peak intensity directly depends on the ?? value as shown on Fig. 8(a).” However, there is not ?? on Fig. 8(a)Author Response
Reviewers 2:
This paper (Sensors-1078234) presents an optical pressure-sensing scheme based on a fiber Bragg grating (FBG) for detecting the transverse compression inside a vitro spinal cord. Due to the inherent property of FBG of high sensitivity to strain measurement, it would be a potential method to investigate the traumatic spinal cord injury (SCI). By utilizing the inherent high-speed response of the Si photodiode-based optical power meter, the response time of the measurement can be improved greatly. The paper gives the fundamental research of the FBG based sensor for traumatic SCI. It is a good exploration by using optical sensors to monitor the real-time biomechanics of soft tissues. The following should be addressed properly before it can be recommended to Sensors.
- The narrow band filter is utilized in Figure 7 to obtain power loss measurements that are correlated to the spectral shift in the FBG’s peak reflectivity. The relationship between the central wavelength of the narrow band filter and reflection spectrum the FBG should be analyzed. To obtain high precision of power loss measurement, the bandwidth of 0.11 nm is relatively large, which should be narrowed further greatly.
Response : The sensor calibration was actually performed with the OSA which has a resolution of 0.05 nm. Further, our results indicate that the narrowband filter’s bandwidth was narrow enough so as to obtain good accuracy. We found that it is the bandwidth of the FBG (0.56nm) that is the main limitation of the accuracy of the sensor as stated in the last paragraph of Section 3.1.
- In Figure 6, optical power is plotted as a function of spinal cord transverse compression for static loading. How about the repeatability?
Response: We agree with the Reviewer that the repeatability was not sufficiently demonstrated. We have actually performed the experiment 10 times, and observed very good repeatability (practically identical results). To reflect this, we have now added error bars in the corresponding Figure 6.
- Figure 3 presents the reflection spectra of the FBG based transverse pressures on the spinal cord surrogate. The unit of Y-axis is dBm and the contrast ratio is only about 0.68 dB. The reflection ratio is low. The unit of dBm should be checked. It may be the absolute value of optical power.
Response : We thank the Reviewer for pointing us to this issue. We have redrawn this plot for the reflection spectra as normalized with the input power in the revised version of the manuscript.
- In page 6, line 177-178, “As the reflection spectrum passes through the 177 narrowband filter, the decrease in peak intensity directly depends on the ?? value as shown on Fig. 8(a).” However, there is not ?? on Fig. 8(a)
Response: We thank the Reviewer for noticing this typographical error. We have made the correction in the revised version of paper.

Reviewer 3 Report
This paper demonstrates an optical pressure-sensing scheme based on FBG sensor, which is embedded in a surrogate spinal cord. The experimental results show that the spatial resolution and time response of the FBG is good enough to be used in real-time in vitro studies of traumatic SCI. But I think more information is need to support the conclusion.
My comments are listed below:
- The spinal cord surrogate was made of silicone elastomer foam. Its shape and dimension were quite similar to human beings thoracic spinal cord. But, when it was applied a force or stress, its mechanical response was determined by the Young's modulus, which is in turn to influence the reflected spectrum of the FBG. So, I suggest that the author could research the mechanical characteristics of the spinal cord surrogate experimentally before use it in optical sensing.
- In Fig.3, the response spectrum of FBG is shown by different pressure. But in the experiments, different weight were used to apply on the spinal cord surrogate. Although in line 118, “the diameter of the different weights applied was kept constant at 1cm”, I still curious about the difference if the diameter was changed. Hope more supplements to be added to make it clear.
Author Response
This paper demonstrates an optical pressure-sensing scheme based on FBG sensor, which is embedded in a surrogate spinal cord. The experimental results show that the spatial resolution and time response of the FBG is good enough to be used in real-time in vitro studies of traumatic SCI. But I think more information is need to support the conclusion.
My comments are listed below:
- The spinal cord surrogate was made of silicone elastomer foam. Its shape and dimension were quite similar to human beings thoracic spinal cord. But, when it was applied a force or stress, its mechanical response was determined by the Young's modulus, which is in turn to influence the reflected spectrum of the FBG. So, I suggest that the author could research the mechanical characteristics of the spinal cord surrogate experimentally before use it in optical sensing.
Response: We agree with the Reviewer that the issue of the validity of the spinal cord surrogate is important. The same point was noted by Reviewer #1 - Comment #4. We kindly refer the reviewers to our answer therein.
- In Fig.3, the response spectrum of FBG is shown by different pressure. But in the experiments, different weight were used to apply on the spinal cord surrogate. Although in line 118, “the diameter of the different weights applied was kept constant at 1cm”, I still curious about the difference if the diameter was changed. Hope more supplements to be added to make it clear.
Response: The mechanical pressure applied on the spinal cord surrogate will change as per the classical relation: Pressure = Force / Area. We thus made sure to keep the diameter of the steel impactor constant while varying the applied Force (via the change in weights) so as to perform a systematic study. We agree that it could be interesting to change slightly the size of the impactor in a future study, but this is outside the scope of the current work.

Reviewer 4 Report
This manuscript details the use of an optical fiber embedded within a silicone elastomer used as a surrogate for a spinal model. The authors generally describe the technique they use to analyze their data and correlate redshifts in their recorded optical power spectrum with radial pressure on their surrogate model. While this concept is interesting, there is quite a bit of characterization of the experimental system as a biologically-relevant model and the accuracy and speed of their measurement technique that must be incorporated into this work before it is publishable. The authors should address the major comments below:
Figure 1: Add something (arrow, etc.) showing direction of the force being applied and something to indicate the coordinate system.
Line 92: Reference [25] is incorrect here. Should it be [15]? Please ensure all of your references are appropriately placed within the text.
Lines 98-100: Please define all symbols. What are epsilon x and epsilon y? I’m assuming strain components, but the authors should make this clear if they are going to discuss these symbols.
Figures 4, 6, and 8: Please show error bars. If tests were repeated multiple times, these data should all be plotted, or at least errors plotted in these figures.
Line 176: How does the error in the fit of ~8.9% affect the conclusions the authors wish to draw from their experimental setup. Please discuss this in your discussion.
Line 188: The power loss units are confusing—please rephrase as something like “relative power loss“ in units of %-1
Line 194: The authors claim their method is accurate. However, the error associated with their fitting is fairly large, and especially with regard to the magnitudes of the strain they wish to observe. This should be incorporated into the discussion. I highly recommend that the authors perform a formal uncertainty analysis and address and characterize the sources of error within the sensor to satisfy this concern and justify their claims of the accuracy of their model setup.
Lines 190-206: The authors discuss the speed of the measurement as a major advantage of their technique however do not present any results showing a time-series recording of the data they gather from their sensor. I would think presenting data recorded using their method during dynamic loading be the most effective way to demonstrate the benefit of their technique. Without this data, I do not think the authors can make claims regarding the sampling rate of their system.
The authors should also discuss the manner in which the material in which the fiber is embedded—the spine surrogate—affects their measurements. For example, how does the compliance of the structure affect the magnitude of the strains reliably measured using their method? Is the fiber strain-coupled to the spine surrogate during the entirety of the experiment and how was this ensured? How is the elastomer foam a relevant spine model and how does its mechanical properties compare with a spine in vivo and ex vivo?
The authors may consider the methods used in the following work to characterize strain measurements made for an optical fiber embedded within a polymer block. While this reference uses optical backscatter reflectometry, rather than FBGs, the mechanical results should be discussed:
- Heinze, Søren, and Andreas T. Echtermeyer. 2018. “A Running Reference Analysis Method to Greatly Improve Optical Backscatter Reflectometry Strain Data from the inside of Hardening and Shrinking Materials.” Applied Sciences (Switzerland) 8(7).
Author Response
This manuscript details the use of an optical fiber embedded within a silicone elastomer used as a surrogate for a spinal model. The authors generally describe the technique they use to analyze their data and correlate redshifts in their recorded optical power spectrum with radial pressure on their surrogate model. While this concept is interesting, there is quite a bit of characterization of the experimental system as a biologically-relevant model and the accuracy and speed of their measurement technique that must be incorporated into this work before it is publishable. The authors should address the major comments below:
Figure 1: Add something (arrow, etc.) showing direction of the force being applied and something to indicate the coordinate system.
Response: We have added a schematic of the opto-mechanical setup in Fig. 1(b) which now clearly shows the direction of the applied force on the spinal cord surrogate.
Line 92: Reference [25] is incorrect here. Should it be [15]? Please ensure all of your references are appropriately placed within the text.
Response: We have checked the corresponding references, and confirm that they are correct.
Lines 98-100: Please define all symbols. What are epsilon x and epsilon y? I’m assuming strain components, but the authors should make this clear if they are going to discuss these symbols.
Response: We elaborated the symbols in the revised manuscript.
Figures 4, 6, and 8: Please show error bars. If tests were repeated multiple times, these data should all be plotted, or at least errors plotted in these figures.
Response: We plotted error bars in the corresponding figures in the revised manuscript.
Line 176: How does the error in the fit of ~8.9% affect the conclusions the authors wish to draw from their experimental setup. Please discuss this in your discussion.
Response: There was a typo in the manuscript, which is taken care of in the improved version of the manuscript. The error appears due to the limitations posed by narrow band pass filter. We have used both OSA and narrow band pass filter in our experimental set-up. The resolution of the narrow band pass filter is five times better than that of the FBG used in our set-up. For even better results (performance) we must use a filter with better resolution, which is not available in our lab.
Line 188: The power loss units are confusing—please rephrase as something like “relative power loss“ in units of %-1
Response: We thank the Reviewer for pointing us to this issue. As indicated to our response to Comment #3 of Reviewer #2, we have corrected the power units in the figure and changed the axis to “Relative power loss”.
Line 194: The authors claim their method is accurate. However, the error associated with their fitting is fairly large, and especially with regard to the magnitudes of the strain they wish to observe. This should be incorporated into the discussion. I highly recommend that the authors perform a formal uncertainty analysis and address and characterize the sources of error within the sensor to satisfy this concern and justify their claims of the accuracy of their model setup.
Response: As per the suggestion of the Reviewer, we have addressed this lack of information on the uncertainty analysis by adding error bars in the figures. We also discussed this issue in our response to Comment on Line 176 above.
Lines 190-206: The authors discuss the speed of the measurement as a major advantage of their technique however do not present any results showing a time-series recording of the data they gather from their sensor. I would think presenting data recorded using their method during dynamic loading be the most effective way to demonstrate the benefit of their technique. Without this data, I do not think the authors can make claims regarding the sampling rate of their system.
Response- We agree that we should have performed dynamic measurements. Our experimental set-up is in a hospital and due to prevalent Covid-19 situation, we are not allowed to access the lab from last nine month. Upon enquiry we have been informed that we will not be allowed in the lab in near future. But we assure the reviewer that once lab entry is granted to us, we will perform the experiment and will publish it in our future article.
- The authors should also discuss the manner in which the material in which the fiber is embedded—the spine surrogate—affects their measurements. For example, how does the compliance of the structure affect the magnitude of the strains reliably measured using their method? Is the fiber strain-coupled to the spine surrogate during the entirety of the experiment and how was this ensured? How is the elastomer foam a relevant spine model and how does its mechanical properties compare with a spine in vivo and ex vivo?
Response: We thank the Reviewer for highlighting important points of discussion. Regarding the selection of the material for producing the spinal cord surrogate and its validity as a proxy for real human spinal cords, we kindly refer the Reviewer to our response to Comment #4 of Reviewer #1. We gently inserted the optical fiber with FBG inside the spinal cord surrogate and made sure that it is at all times free to move without restraint; so it was not strain-coupled.
- The authors may consider the methods used in the following work to characterize strain measurements made for an optical fiber embedded within a polymer block. While this reference uses optical backscatter reflectometry, rather than FBGs, the mechanical results should be discussed:
- Heinze, Søren, and Andreas T. Echtermeyer. 2018. “A Running Reference Analysis Method to Greatly Improve Optical Backscatter Reflectometry Strain Data from the inside of Hardening and Shrinking Materials.” Applied Sciences (Switzerland) 8(7).
Response: We thank the Reviewer for pointing to us this relevant paper. We have added a short discussion in the introduction and cited the corresponding reference above: “”

Round 2
Reviewer 2 Report
The authors should reply to the reviewer's comments one by one in a reponse file.
Author Response
Reviewers 2:
This paper (Sensors-1078234) presents an optical pressure-sensing scheme based on a fiber Bragg grating (FBG) for detecting the transverse compression inside a vitro spinal cord. Due to the inherent property of FBG of high sensitivity to strain measurement, it would be a potential method to investigate the traumatic spinal cord injury (SCI). By utilizing the inherent high-speed response of the Si photodiode-based optical power meter, the response time of the measurement can be improved greatly. The paper gives the fundamental research of the FBG based sensor for traumatic SCI. It is a good exploration by using optical sensors to monitor the real-time biomechanics of soft tissues. The following should be addressed properly before it can be recommended to Sensors.
- The narrow band filter is utilized in Figure 7 to obtain power loss measurements that are correlated to the spectral shift in the FBG’s peak reflectivity. The relationship between the central wavelength of the narrow band filter and reflection spectrum the FBG should be analyzed. To obtain high precision of power loss measurement, the bandwidth of 0.11 nm is relatively large, which should be narrowed further greatly.
Response : The sensor calibration was actually performed with the OSA which has a resolution of 0.05 nm. Further, our results indicate that the narrowband filter’s bandwidth was narrow enough so as to obtain good accuracy. We found that it is the bandwidth of the FBG (0.56nm) that is the main limitation of the accuracy of the sensor as stated in the last paragraph of Section 3.1.
- In Figure 6, optical power is plotted as a function of spinal cord transverse compression for static loading. How about the repeatability?
Response: We agree with the Reviewer that the repeatability was not sufficiently demonstrated. We have actually performed the experiment 10 times, and observed very good repeatability (practically identical results). To reflect this, we have now added error bars in the corresponding Figure 6.
- Figure 3 presents the reflection spectra of the FBG based transverse pressures on the spinal cord surrogate. The unit of Y-axis is dBm and the contrast ratio is only about 0.68 dB. The reflection ratio is low. The unit of dBm should be checked. It may be the absolute value of optical power.
Response : We thank the Reviewer for pointing us to this issue. We have redrawn this plot for the reflection spectra as normalized with the input power in the revised version of the manuscript.
- In page 6, line 177-178, “As the reflection spectrum passes through the 177 narrowband filter, the decrease in peak intensity directly depends on the ?? value as shown on Fig. 8(a).” However, there is not ?? on Fig. 8(a)
Response: We thank the Reviewer for noticing this typographical error. We have made the correction in the revised version of paper.

Reviewer 4 Report
The manuscript is much improved by the author's revisions. I recommend accepting this manuscript for publication. Great work! However, figure 8 is referenced before figure 6, and I recommend renumbering the figures by the order in which they appear in the text prior to publication.
Author Response
Reviewer 4:
The manuscript is much improved by the author's revisions. I recommend accepting this manuscript for publication. Great work! However, figure 8 is referenced before figure 6, and I recommend renumbering the figures by the order in which they appear in the text prior to publication.
Response: Authors thanks to reviewers for appreciating our work. Authors also thankful for reviewers for additional comments.
We also removed typo error in the line 147. Also, authors checked the numbering of figure.
